

# An adaptive fusion-based data augmentation method for abstract dialogue summarization

Weihao Li, Dan Jiang, Han Zhang, Kejing Xiao and Shaozhong Cao

School of Information Engineering, Beijing Institute of Graphic Communication, Beijing, China

## ABSTRACT

The dialogue summarization is necessary for information retrieval, and the training of abstract dialogue summarization models heavily rely on large amounts of labeled data. However, manual summarization of long dialogue is labor-costing and time-consuming. To solve this problem, this article proposes a data augmentation method for dialogue summary based on adaptive augmentation fusion (AAF), integrating the strengths of both Minor Perturbation Augmentation (MPA) and Semantic Reconstructive Augmentation (SRA) to balance model learning effectiveness and generalization capabilities. We first integrated existing enhancement methods to address the problem of insufficient annotated data for dialogue summarization. The experimental results on both the DialogSum and SAMSum datasets demonstrate that the AAF method achieves significant improvements in ROUGE scores under resource-constrained conditions, outperforming baseline approaches. Furthermore, it was validated that the selection of the amount of augmented data has a significant impact on model training results under resource-constrained conditions. We have publically released our code at https://github.com/alolke/AAF.

## INTRODUCTION

With the widespread use of social media in daily life and work, the need for instant and efficient access to dialogue information has become both important and pressing. This highlights the significance of dialogue summarization technology, which markedly enhances the efficiency of information transmission. However, current abstract dialogue summarization techniques heavily rely on large amounts of labeled data. In real-world settings, dialogues are complex, and manual summarization is costly, making labeled data acquisition highly challenging. Therefore, improving the generation capabilities for abstract dialogue summaries has become an urgent need. Current research methods for abstract dialogue summarization are diverse, encompassing multi-level conversation modeling (*Zhao et al., 2019*; *Zhu et al., 2020*), dialogue act utilization (*Goo & Chen, 2018*), key phrase and entity extraction (*Narayan et al., 2021*), topic segmentation (*Liu et al., 2019*; *Sahu & Laradji, 2024*), word graph degradation (*Park & Lee, 2022*), conversation relationship extraction (*Chen & Yang, 2021b*), and dynamic graph-based knowledge

Corresponding author
Dan Jiang, jiangdan@bigc.edu.cn

aggregation. Most of these summarization approaches, however, rely heavily on large amounts of labeled data to train models.

In reality, resources for dialogue summarization data are quite limited. For example, the SAMSum dataset has only 16,369 summaries, and the more recent DialogSum dataset offers just 12,460 samples—significantly fewer than other text-based tasks. While these numbers may seem moderate in absolute terms, they are insufficient for training robust dialogue summarization models due to the inherent complexity of the task. Compared to simpler tasks such as text classification or sentiment analysis, which can achieve reasonable performance with tens of thousands of samples, dialogue summarization requires understanding multi-turn conversations, speaker intentions, and informal language patterns, which demand a much more diverse and extensive training set. Furthermore, unlike machine translation tasks that benefit from large-scale parallel corpora (*e.g.*, WMT datasets with millions of sentence pairs), dialogue summarization lacks such abundant resources, making data augmentation a critical strategy to bridge this gap. Compared to large-scale pretraining tasks (*e.g.*, BERT or GPT) that utilize millions of samples, the limited size of SAMSum and DialogSum highlights the need for data augmentation to artificially expand the dataset and improve model generalization.

On one hand, manually annotating high-quality dialogue summary data is a time-consuming, labor-intensive process. On the other hand, despite the abundance of conversational data available, it is often lengthy and requires complex preprocessing and truncation to reduce noise. Additionally, daily dialogues and chat logs are often difficult to comprehensively cover all aspects of life, increasing the risk of model overfitting. To address these challenges, data augmentation techniques can play a critical role by generating diverse and realistic dialogue samples, thereby improving the robustness and generalization capabilities of summarization models. Previous studies have demonstrated that data augmentation can significantly enhance performance in low-resource summarization tasks (cite relevant work), further justifying its importance in our approach.

To address the problem of data scarcity, data augmentation techniques are viewed as an effective solution. However, existing data augmentation methods have limited effectiveness in augmentation performance under resource constraints, particularly when generating augmented data while inadvertently introducing uncontrollable noise. Overly strict noise control can confine the results to the original data, preventing the effective incorporation of diverse information and thereby restricting the model's generalization capability. Consequently, a key challenge that needs to be resolved is how to reasonably introduce diverse information while maintaining the fidelity of the augmented data.

In light of these challenges, this article proposes a novel data augmentation method for dialogue summary, referred to as Adaptive Augmentation Fusion (AAF). This method combines the advantages of Minor Perturbation Augmentation (MPA) and Semantic Reconstructive Augmentation (SRA), aiming to adaptively balance the model's learning effectiveness and generalization capabilities when utilizing augmented data. In terms of specific implementation, the innovative approach categorizes data augmentation methods into two types—MPA and SRA—based on different levels of semantic intervention in the

text. In the method of AAF, the proportions of MPA and SRA are dynamically adjusted based on the characteristics of the data and the requirements of the model. This enables the integration of various data augmentation methods within the original dataset.

The key contributions of this article are as follows:

- This article presents a data augmentation method for dialogue summary based on AAF. During the augmentation process, the proportions of different augmentation methods are adaptively adjusted to achieve a balance between model learning effectiveness and generalization capabilities. Unlike traditional static augmentation approaches, our method leverages real-time validation performance as feedback to guide the augmentation strategy, ensuring that the augmented data remains aligned with the model's learning progress.
- We define an adaptive augmentation fusion strategy that utilizes validation results during the training process as evaluation metrics to dynamically adjust the fusion ratios. Furthermore, we provide a comprehensive theoretical analysis of how the AAF strategy influences model learning, including its impact on gradient optimization and feature representation. This analysis offers insights into the underlying mechanisms that make adaptive augmentation effective for dialogue summarization tasks.
- The effectiveness of the augmentation method is validated on two major dialogue summary datasets, DialogSum and SAMSum. The experimental results demonstrate significant improvements in ROUGE scores (ROUGE-1, ROUGE-2, and ROUGE-L) compared to baseline models, highlighting the superiority of our adaptive data augmentation approach.

## LITERATURE REVIEW

### Abstract dialogue summarization

Abstract dialogue summarization (*Shafiq et al., 2023*) is a key information extraction technique that presents the core information of dialogue content to enhance the efficiency of information transmission. This technique finds widespread application across various fields, including meetings, doctor-patient communication, customer service dialogues, debates, interviews, and daily conversations.

Recent research on abstract dialogue summarization has focused on various aspects of conversation summarization. For instance, *Feng et al. (2021)* employed DialoGPT to predict unpredictable words as keywords and incorporated these into the summarization model to capture more informative content. *Lei et al. (2021a)* proposed a hierarchical transformer-based dialogue summarization model that differentiates the relationship between speakers and their corresponding pronouns. *Kim et al. (2022)* introduced the Summarizing with Injected Commonsense Knowledge (SICK) framework, which utilizes common sense reasoning as additional context to incorporate common knowledge—such as intentions and reactions—during the dialogue summarization process, aiding in the capture of implicit meanings and emotions in conversations. Additionally, *Murray, Renals & Carletta (2005)* and *Zechner (2002)* utilized term frequency–inverse document

frequency (TF-IDF) to compute sentence vectors, identifying similar or redundant sentences by calculating the cosine similarity between these vectors. Such similar sentences were deemed redundant and removed, while the remaining sentences were retained for summarization. *Lei et al. (2021b)* represented the themes in conversations as a graph structure to capture thematic connections and hierarchies, using ConceptNet to find related terms within the dialogue for a more flexible thematic structure. The comparison of the advantages and disadvantages of related literature is shown in Table 1.

However, the aforementioned methods still rely heavily on large amounts of high-quality labeled data. Acquiring a high-quality dataset is often a time-consuming and labor-intensive process. A high-quality dialogue dataset not only needs to cover a wide range of conversational domains but should also include various types of dialogue patterns to ensure data richness and practicality. Consequently, employing data augmentation methods to generate more diverse data holds promise for alleviating the current data scarcity problem.

## Data augmentation in NLP

In the field of natural language processing (NLP), data augmentation techniques are widely employed to alleviate the problem of data scarcity and enhance the generalization capabilities of models. These techniques generate additional augmented data to help models perform more robustly across various tasks. Data augmentation methods are classified into token-level augmentation, sentence-level augmentation, segmented based augmentation adversarial data augmentation, and hidden space augmentation (*Chen et al., 2023*; *Ziyaden et al., 2024*).

**Token-level augmentation** processes words and phrases within sentences to generate augmented text, ideally preserving the semantics and labels of the original text. Common techniques include operations such as insertion, deletion, and swapping of words or phrases (*Wei & Zou, 2019*), generating new data that is semantically similar yet distinct from the original text. This may involve synonym replacement using predefined dictionaries like WordNet (*Kolomiyets, Bethard & Moens, 2011*) or leveraging similarity in word embedding spaces (*Wang & Yang, 2015*). Recent research has also focused on compositional augmentation (*Andreas, 2019*), which recombines different fragments from various sentences to create enhanced examples.

**Sentence-level augmentation** modifies entire sentences. *Zhang, Yang & Yang (2022)* utilized the hierarchical structure of sentences, employing constituency parsing to decompose sentences into meaningful substructures and recombine them to generate new training samples. *Anaby-Tavor et al. (2020)* generated new text conditionally based on labels using language models.

**Segmented based augmentation** is a data augmentation technique that modifies text at the segment level rather than individual tokens. It involves operations such as segment reordering, replacement, or paraphrasing to generate diverse yet semantically coherent training data. This method is particularly effective for tasks like dialogue summarization, where preserving discourse structure and context is critical. For example, *Ouyang et al. (2023)* used this method. Its key advantage lies in producing linguistically plausible

**Table 1 Advantages and disadvantages of recent approaches in abstract dialogue summarization.**

| Literature reference | Advantages | Disadvantages |
|---|---|---|
| *Feng et al. (2021)* | 1. Uses DialoGPT to predict keywords, enhancing the informativeness of summaries.<br>2. Capable of capturing key information in dialogues. | 1. Relies on the performance of the pre-trained model DialoGPT.<br>2. Dependence on unpredictable words may lead to instability in summaries. |
| *Lei et al. (2021a)* | The hierarchical Transformer model can distinguish the relationship between speakers and pronouns. Improves the understanding of dialogue structure. | High model complexity and training costs. Limited capability in handling long dialogues. |
| *Kim et al. (2022)* | 1. Introduces the SICK framework, enhancing the capture of implicit meanings in dialogues.<br>2. Capable of handling emotions and intentions in dialogues. | 1. The inclusion of commonsense reasoning increases computational complexity.<br>2. Dependence on commonsense knowledge bases may reduce domain adaptability. |
| *Murray, Renals & Carletta (2005)* | 1. Uses TF-IDF and cosine similarity to remove redundant sentences, simplifying summaries.<br>2. The method is simple and easy to implement. | Relies solely on word frequency statistics, potentially ignoring semantic information. Poor performance on short texts. |
| *Lei et al. (2021b)* | 1. Represents dialogue themes as a graph structure, capturing thematic hierarchies.<br>2. Introduces ConceptNet to enhance thematic relevance. | High complexity in constructing and optimizing graph structures. |

variations, enhancing model generalization. However, it requires accurate segmentation tools and careful control to avoid semantic inconsistencies.

**Adversarial data augmentation** affects model predictions and confidence by introducing adversarial perturbations to the original data without altering human judgments. *Zhu et al. (2019)* directly incorporated adversarial perturbations into labeled embeddings or sentence hidden representations to produce new data.

**Hidden-space augmentation** intervenes within the feature space generated by the model to create new data samples. *Miao et al. (2020)* performed interpolation between two or more data points, while *Chen et al. (2021a)* dynamically and strategically removed continuous spans within the hidden space to encourage the model to learn more generalizable features.

This article introduces a new classification of data augmentation methods tailored for dialogue texts, specifically considering the strong coherence in dialogue structures. Unlike the aforementioned classification that emphasizes the operational implementation, we propose a classification that categorizes methods into MPA and SRA based on their effect on inter-sentence coherence. MPA involves slight modifications to words or meanings, maintaining the main content and structure. Conversely, SRA involves extensive semantic adjustments or reconstructions of the text, introducing a greater diversity of augmented samples.

# METHODS

To address the problem of data scarcity in the abstract dialogue summary task, as well as the challenge of effectively incorporating diverse information into data augmentation, this

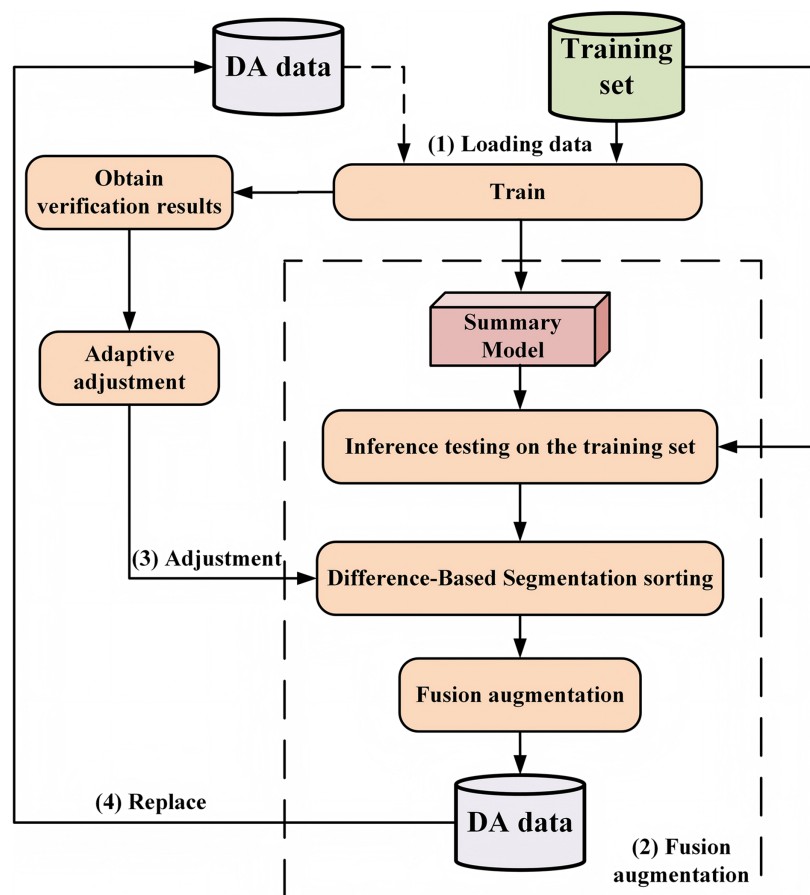

**Figure 1 Method of adaptive augmentation fusion (AAF), which includes: (1) model loading training data.** At the beginning of training, only the initial training model is loaded; after the second adjustment, a mixture of initial training data and augmented data is loaded. (2) Fusion augmentation process. (3) Adaptive parameter adjustment. (4) Replacement of augmented data after each adjustment.

section proposes a data augmentation method for dialogue summary based on AAF. This method synthesizes the characteristics of both MPA and SRA. The specific methodological workflow is illustrated in Fig. 1.

## Fusion augmentation

In data augmentation, the generation of augmented data typically has two main objectives: to reinforce the model's learning of problem-solving patterns in the original dataset and to improve the diversity of the dataset. However, research has shown that these two objectives can be contradictory during the generation process, particularly evident in the dialogue summarization task where data scarcity is a significant problem. MPA, which involves slight modifications such as synonym replacement or minor syntactic adjustments, can maintain the original data patterns to a certain extent, allowing the model to learn features from the data more deeply. This approach, however, can lead to model overfitting on the training data, which may impair its generalization capacity. By contrast, SRA introduces greater variability by generating diverse sentence structures, thereby encouraging the

model to learn a broader range of features and enhancing its generalization capabilities. Yet, when the proportion of diversified augmented data is too high, the model can become susceptible to noise interference. Unlike MPA, SRA may cause the model to overly rely on the newly generated data patterns, thereby reduceing its learning effectiveness regarding the original data.

Despite the inherent contradictions between MPA and SRA, they exhibit significant complementarity in the model learning process. These two augmentation techniques expand the data from both micro and macro perspectives. MPA functions as a micro-level augmentation, operating as a local search within the input space based on patterns the model has already learned, allowing it to fine-tune its understanding of specific features. This resembles a local convergence process in gradient descent, where the model optimizes the patterns it has internalized. In contrast, SRA serves as a macro-level augmentation, exploring a wider input space to expose the model to varied semantic expressions, thereby enhancing its ability to capture deeper, underlying patterns within the data.

Therefore, AAF naturally integrates these two types of augmentation methods to leverage their complementarity fully. This approach can enrich the data patterns while enhancing the model's learning and generalization capabilities. Specifically, the part of fusion augmentation consists of three main components: Minor Perturbation Augmentation, Semantic Reconstructive Augmentation, and the Fusion Augmentation Method, as illustrated in Fig. 2.

### Minor perturbation augmentation

We are considering using the MPA strategy for samples where the model's performance in summary generation is declining during current training. These samples reflect patterns that the model has not fully mastered or exhibits instability in performance. Therefore, there is no need to introduce overly complex changes. The core objective of MPA is to solidify the model's understanding of existing data patterns while avoiding unnecessary noise risks caused by overly complex modifications.

Specific examples of MPA methods include:

**Synonym Replacement (SR):** This involves replacing certain words in the samples with their synonyms without altering the core meaning. This ensures that the model maintains robust performance under slightly modified inputs. Given the structural characteristics of long dialog texts, random replacements are not permitted; instead, replacements are concentrated on specific key sentences or paragraphs, with minimal changes to other parts.

**Utterance Random Exchange (UR)** (*Vlachos, Stafylakis & Androutsopoulos, 2024*): This method disrupts the relationships between utterances to create augmented dialogues. It first randomly selects two utterances in the dialogue and then swaps them.

**Back Translation (BT)** (*Xie et al., 2020*; *McNamee & Duh, 2023*): This approach translates the selected utterance into an intermediate language before translating it back into the original language. It serves as a sentence-level data augmentation method that effectively utilizes the differences across various linguistic cultures, yielding a broader range of language patterns and structural variants to enhance the model's generalization capability.

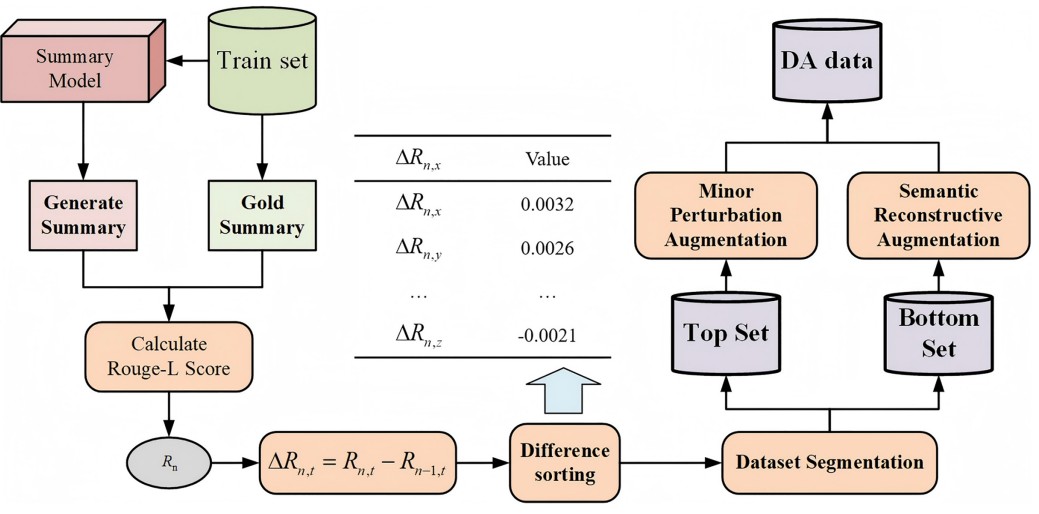

**Figure 2 Augmentation fusion process diagram.**

**Action Behavior Insertion** (*Chen & Yang, 2021a*): By leveraging the unique characteristics of dialogues, such as interruptions, repetitions, false starts, and confirmations, the dialogue-action-guided insertion method is employed to disrupt dialogues and produce augmented data.

Through these simple modifications, the model can further reinforce its learning based on known patterns without introducing excessive new information. This approach ensures that the model consolidates its existing knowledge under slightly modified inputs, enhancing its understanding and adaptability to the original patterns.

### Semantic reconstructive augmentation

For data samples that demonstrate improved performance, we believe that the training model has sufficiently understood and learned the information contained within them; thus, we employ the SRA strategy. Since these samples have already achieved good outputs in previous training, introducing greater diversity aids the model in further expanding its feature space and deepening its understanding and adaptability to different types of data.

Specific implementations of SRA include:

**Contextual Information Incorporation:** This involves reconstructing sentences by introducing relevant background information or context, thereby altering their overall meaning.

**Semantic Extraction and Rewriting** (*Min et al., 2023*): This method extracts key information or themes from sentences and rewrites them based on the extracted information, expressing the same semantics in different ways.

**Adversarial Generation** (*Liu & Sun, 2023*): Utilizing generative adversarial networks (GANs), this approach creates new sentences that result in substantial differences from the original sentences.

**Text Generation** (*Kumar, Choudhary & Cho, 2020*): This method employs pretrained language models (*e.g.*, GPT, T5) to generate new sentences or paragraphs, potentially altering the structure and expression of the original text significantly.

These methods enable the model to incorporate more diverse data patterns and information, thereby enhancing its generalization capability and making it more effective at adapting to previously unseen complex samples.

### Fusion augmentation method

In the method of fusion augmentation, we implement precise augmentation for different data samples. Existing augmentation methods typically select augmented data targets based on the model's performance on the dataset, primarily focusing on samples with poorer outcomes for data augmentation. An excessive focus on poorly performing data may lead the model to overlook or weaken its learning of good samples, resulting in an imbalanced learning process.

In this article, we propose a new approach—not only to augment poor-performing data but also to appropriately augment well-performing data. We categorize the data into two subsets, "good" and "bad" based on the model's learning outcomes, and employ different augmentation methods for these subsets.

The specific process for dataset partitioning is as follows: after completing the $n$-th iteration of training, the trained summarization model is tested on the training set and the Rouge score between each generated summary, and the gold standard summary is calculated. Subsequently, the Rouge-L value from the current iteration is subtracted from the Rouge-L value from the previous iteration to determine the trend in changes of the text summary's Rouge-L score, as shown in the following formula:

$$\Delta R_{n,t} = R_{n,t} - R_{n-1,t}. \tag{1}$$

Here, $R_{n,t}$ represents the Rouge-L score between the summary generated by the trained model for the $t$-th data in the training set and the gold standard summary during the $n$-th iteration. $R_{n-1,t}$ denotes the Rouge-L score for the same $t$-th data in the training set during the $n-1$-th iteration. Let $\Delta R_{n,t}$ be the difference in Rouge-L scores between the two iterations.

The difference in Rouge-L scores is used to analyze the model's understanding of the training set data during that iteration. According to the difference, classify the data into "good" and "bad" data, referred to as the Top Set and Bottom Set, respectively. The segmentation ratio will be adjusted according to the augmented fusion coefficient.

In the experiments, the data in the Bottom Set represents the model's current weak points in learning. Applying MPA to this portion allows the model to continue strengthening its learning on samples where its performance is suboptimal, ensuring more robust performance on these samples. Conversely, SRA is used to augment the data in the Top Set, challenging the model's learning capabilities and improving its performance in handling various complex scenarios while reducing the risk of overfitting. Different augmentation strategies are applied to these two data types. Through the method of fusion augmentation, the proportions of different augmented data are adaptively adjusted, ensuring that the model can better adapt to complex tasks.

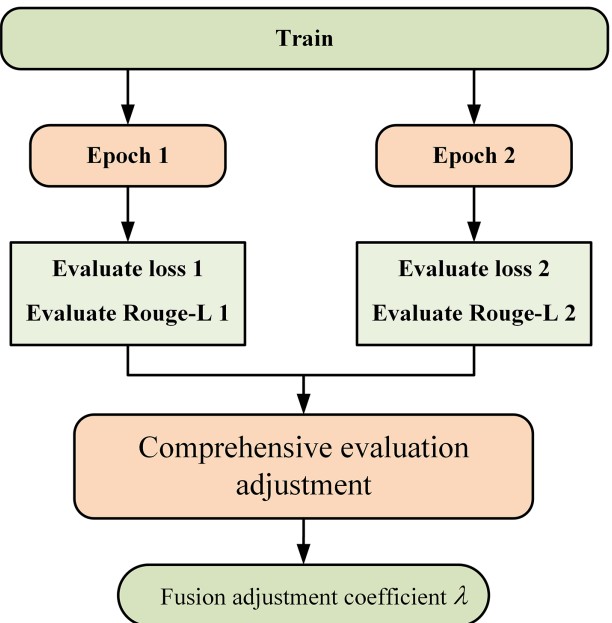

**Figure 3 Adjustment of parameters in adaptive augmentation fusion.**

### Adaptive augmentation fusion strategy

*Initialization fusion strategy*

At the beginning of the training, the proposed method initializes the fusion ratio of MPA and SRA, allowing them to generate 50% of the data for the training set, respectively. Therefore, the initial fusion coefficients $\omega$ for both MPA and SRA are set to 0.5, with their sum equal to 1. To further optimize the fusion ratio, we introduce a fusion adjustment coefficient $\lambda$, with an initial value set to 0. As training progresses, this coefficient is dynamically adjusted based on the model's performance, with the adjustment amount denoted as $\Delta\lambda$. The calculation of the augmented fusion coefficients is performed using the following formula:

$$\lambda_n = \lambda_{n-1} + \Delta\lambda \tag{2}$$
$$\omega_P' = \omega_P - \lambda_n \tag{3}$$
$$\omega_R' = \omega_R - \lambda_n. \tag{4}$$

Here, $\omega_P$ and $\omega_R$ represent the fusion coefficients for MPA and SRA, respectively; $\omega_P'$ and $\omega_R'$ are the adjusted coefficients. $\lambda_n$ denotes the adjustment coefficient for the $n$-th round of training.

*Define the adjustment criteria and augmentation strategies for the fusion adjustment coefficient*

Figure 3 illustrates the coefficient adjustment process. During the training process, the fusion adjustment coefficient is changed every two epochs. After training for two epochs, the model yields two validation results, which include the evaluate loss value and the

**Table 2 Adjustment criteria for the fusion adjustment coefficient.**

| Trend of evaluate loss | Trend of evaluate Rouge-L | Fusion adjustment coefficient | Operation |
|---|---|---|---|
| Decreased (↓) | Increased (↑) | $\lambda + \Delta\lambda$ | Increase the proportion of SRA |
| Increased (↑) | Increased (↑) | $\lambda - \Delta\lambda$ | Increase the proportion of MPA |
| Decreased (↓) | Decreased (↓) | $\lambda - \Delta\lambda$ | Increase the proportion of MPA |
| Increased (↑) | Increased (↓) | — | Overfitting, stop training |

evaluate Rouge-L value. The fluctuations in the loss values and Rouge-L values from the two validation results serve as the criteria for adjusting the augmentation strategies. The criteria are presented in Table 2.

There are four possible scenarios when comparing the Rouge values and loss values:

An ideal scenario occurs when the Rouge values continuously improve while the loss values decrease. This indicates that the model is optimizing, the quality of the generated text is improving, the introduction of noise remains within a controllable range, and overfitting has not occurred. Therefore, more SRA data should be added to enhance the model's generalization capacity, enabling it to better adapt to previously unseen complex samples.

If the Rouge values increase but the loss values also rise, this suggests that the quality of the model's generation is improving, but there may be instability during the training process. Conversely, if the Rouge values decrease alongside the loss values, this indicates that the model has learned features during training but has failed to effectively enhance the generation quality. These two scenarios may arise from the presence of noise or bias in the training data, leading to fluctuations in learning. Consequently, we aim to enhance the data to align more closely with the original data, thus introducing MPA data to reinforce the existing generation patterns.

The least favorable scenario occurs when both the Rouge values and loss values deteriorate. This usually signifies significant problems during the model training process. The training data may be of poor quality, containing substantial noise or experiencing overfitting, making it difficult for the model to learn effective features. Therefore, training should be halted. In the augmentation strategy, it is necessary to adhere to a gradual adjustment to avoid severe fluctuations that may cause the model to be unable to adapt. The selection of adjustment range was experimentally verified in "Literature Review", and it was found that maintaining an adjustment range of 0.1 is better. Ultimately, the augmented fusion coefficient for either of the two augmentation methods will not be less than 0.3 and will not exceed 0.7.

## EXPERIMENT

### Dataset

This study employs two widely-used dialogue summarization datasets, SAMSum (*Gliwa et al., 2019*) and DialogSum (*Chen et al., 2021b*), to validate the effectiveness of the proposed dialogue data augmentation method. Both datasets are English-based and consist of dialogue texts paired with corresponding summaries, covering a diverse range of

**Table 3 Dataset statistics.**

| Dataset | Spilt | Number of turns | | | Reference length | | |
|---|---|---|---|---|---|---|---|
| | | Mean | Std | Interval | Mean | Std | Interval |
| SAMSum | Train 14,732 | 11.17 | 6.45 | [1,46] | 23.44 | 12.72 | [2,73] |
| | Dev 818 | 10.83 | 6.37 | [3,30] | 23.42 | 12.71 | [4,68] |
| | Test 819 | 11.25 | 6.35 | [3,30] | 23.12 | 12.20 | [4,71] |
| DialogSum | Train 12,460 | 9.49 | 4.16 | [2,65] | 22.87 | 10.71 | [5,153] |
| | Dev 500 | 9.38 | 3.99 | [2,29] | 20.91 | 9.76 | [6,56] |
| | Test 500 | 9.71 | 4.99 | [2,65] | 19.09 | 9.20 | [6,84] |

everyday conversations and real-life scenarios in open domains. The selection of these datasets is motivated by their complementary characteristics and their established role as benchmarks in dialogue summarization research.

The SAMSum contains 14,732 training samples, 818 validation samples, and 819 test samples, focusing primarily on informal conversations extracted from social media platforms. In contrast, the DialogSum comprises 12,460 training samples, 500 validation samples, and 500 test samples, encompassing a broader spectrum of dialogue types, including both casual and formal conversations. This diversity in dialogue styles and contexts allows for a comprehensive evaluation of the proposed method's generalization capabilities across different conversational settings.

The choice of these two datasets is further justified by their widespread adoption in the literature, which facilitates direct comparison with state-of-the-art methods. Additionally, utilizing multiple datasets helps mitigate potential biases inherent in single-dataset evaluations, thereby enhancing the robustness and reliability of the experimental results. In the initial stage of the study, we also reviewed other potential datasets, such as MediaSum and AMI, but they were excluded due to their domain specificity (*e.g.*, news or meeting transcripts) and the fact that the target task does not focus on general-purpose dialogue summarization. Detailed statistics and characteristics of the datasets are summarized in Table 3.

## Baseline

This study compares the AAF with several state-of-the-art techniques and baseline models:

**BART** is the most advanced pre-trained model for summarization tasks, proposed by *Lewis (2019)*. In this experiment, BART-base is utilized as the baseline model for all augmentation methods.

**Synonym Replacement (SR)** (*Khan et al., 2022*) is a character-level data augmentation method that maintains semantic meaning by randomly replacing words in the dialogue with their synonyms.

**Back-translation (BT)** is a conversation-level augmentation method that involves first translating the selected text into an intermediate language and then translating it back to the original language.

**CODA** (*Chen & Yang, 2021a*) proposes a simple yet effective approach for conversation data augmentation in abstract dialogue summarization. The specific operations include random swapping/deletion to disrupt the internal discourse relationships of the conversation, dialogue act-guided insertion to interrupt the flow of the conversation, and condition-based generation for substitutions.

**Compo** (*Ouyang et al., 2023*) is a substructure-level synthetic data augmentation method. It begins by extracting dialogue structures (such as topic splits and action triplets) as fundamental units. These semantically meaningful dialogue segments are then combined to create new training data instances.

## Evaluation metrics

The evaluation metric used in this study is the ROUGE score (*Lin, 2004*). The ROUGE score primarily measures the lexical, phrase, or sentence-level similarity between the generated text and the target reference text.

In this experiment, the ROUGE score is used for different evaluations in two distinct parts. Specifically, during data segmentation, the change in the Rouge-L value is employed to determine the model's learning status on each training data instance. In the adaptive adjustment phase, the Rouge-L value is utilized to assess the overall learning progress of the model during training, alongside the loss value, serving as a reference for determining changes in the adaptive adjustment coefficient.

In this study, the F1 score of the ROUGE metric is generally used as the result, and the formula for calculating the ROUGE score is as follows:

$$P = \frac{LCS(X, Y)}{|Y|} \tag{5}$$

$$R = \frac{LCS(X, Y)}{|X|} \tag{6}$$

$$F1 = \frac{2 \cdot P \cdot R}{P + R}. \tag{7}$$

## Experimental parameters

During the training process, the BART-base model is used to initialize both the encoder and decoder. In the experiment, SR and Compo were used as the basic augmentation methods for AAF. The fusion coefficient undergoes 15 iterations, with each iteration comprising two epochs, a batch size of 32, and a learning rate of $3 \times 10^{-5}$. The fusion coefficient is established at 0.1, and the usage ratio of the augmented data is set at 50%. In the experiments, 1% of each dataset was used as training data, with the quantity denoted as $x$. Fusion augmentation was applied to each training sample once, resulting in an augmented dataset with a final size of $2x$. To ensure a consistent data volume for each training round, the newly generated augmented data replaces the previous augmented data after each iteration, forming a training set composed of both the original and augmented data. To validate the effectiveness of this method under data scarcity conditions, 1% of the dataset is selected as the original training set, and

**Table 4 Comparison of the AAF with other augmentation methods.**

| Dataset | Model | ROUGE-1 | ROUGE-2 | ROUGE-L |
|---|---|---|---|---|
| DialogSum | $BART_{base}$ | 38.98 | 12.62 | 31.05 |
| | SR | 39.91 | 13.84 | 31.86 |
| | BT | 40.76 | 14.63 | 32.42 |
| | CODA | 41.03 | 14.76 | 32.78 |
| | Compo | 40.95 | 14.78 | 32.53 |
| | $AF_{fixed}$ | 39.87 | 13.94 | 31.81 |
| | AAF | **41.38** | **15.09** | **33.52** |
| SAMSum | $BART_{base}$ | 42.82 | 18.69 | 34.84 |
| | SR | 43.38 | **19.16** | 35.36 |
| | BT | 43.22 | 18.71 | 35.01 |
| | CODA | 43.23 | 19.02 | 35.12 |
| | Compo | 43.57 | 18.83 | 35.22 |
| | $AF_{fixed}$ | 43.47 | 18.78 | 35.34 |
| | AAF | **44.57** | 19.04 | **35.65** |

**Note:**
The best result is bolded, and the second best result is underlined.

the average results from three experiments conducted under a random seed are reported as the final outcomes.

## Result

Table 4 presents the performance of various data augmentation methods for the dialogue summarization task on the DialogSum and SAMSum benchmark datasets, under resource-constrained conditions (1% of the number of datasets). The experimental results are measured using ROUGE scores. The AAF is compared with the baseline model BART-base (without data augmentation) and several commonly used data augmentation methods, including Synonym Replacement (SR), Back Translation (BT), the simple augmentation method (CODA) that disrupts internal discourse relationships within conversations, a combination augmentation method based on topic segmentation and action dialogue generation (Compo), and a fixed-ratio augmented fusion method ($AF_{fixed}$). The experimental results indicate that AAF consistently demonstrates superior summarization performance compared to other methods across both datasets.

## Result analysis

As the baseline model for the dialogue summarization task, BART-base demonstrates relatively weak performance, particularly in the ROUGE-2 and ROUGE-3 metrics, where it lags significantly behind the effectiveness of using augmentation methods. This indicates that the baseline model's generalization ability on small-scale datasets is limited, making it difficult to capture the key content within conversations.

Among the individual augmentation methods, SR demonstrated only marginal improvements across both datasets. BT slightly outperformed SR on both datasets, showing significant enhancements in the ROUGE-2 and ROUGE-L metrics; however, the

**Table 5  BERTScore metric on the DialogSum dataset.** The bold values indicate the highest BERTScore metrics achieved in the experiment.

| Model | P | R | F1 |
|---|---|---|---|
| $BART_{base}$ | 0.84 | 0.82 | 0.83 |
| SR | 0.85 | 0.82 | 0.83 |
| BT | 0.88 | 0.87 | 0.88 |
| Compo | 0.89 | 0.89 | 0.90 |
| AAF | **0.91** | **0.92** | **0.91** |

effectiveness of this method in handling complex dialogues did not meet expectations. The CODA method achieved substantial gains on the DialogSum dataset by employing several simple augmentation techniques that disrupt the semantic relationships between conversations. Compo, through topic segmentation and action dialogue generation, balanced the enhancement of dialogue summarization performance, particularly demonstrating superior results in ROUGE-2 and ROUGE-3.

In contrast, the fixed ratio additive fusion (AF) exhibited slight improvements on certain metrics but lacked stability in data performance, particularly when compared to AAF, which showed a deficiency in flexibility. AAF demonstrated the best performance across both datasets. On the DialogSum, AAF achieved ROUGE-1, ROUGE-2, and ROUGE-3 scores of 41.38, 15.09, and 33.52, respectively; while on the SAMSum, the scores were 44.57, 19.04, and 35.65, respectively. Compared to other augmentation methods, AAF better balanced the model's learning and generalization capabilities, demonstrating that adaptive adjustment of the augmentation coefficient effectively mitigates overfitting during training while avoiding the introduction of excessive noisy data.

To further evaluate the quality of the text generated by the model from a semantic perspective, we introduced the BERTScore evaluation metric based on pre-trained language models. Compared to the traditional ROUGE metric, BERTScore calculates the cosine similarity between the generated text and the reference text in the BERT embedding space, enabling a more accurate capture of the deep semantic information of the text. We conducted supplementary experiments on the DialogSum dataset, using BERTScore to evaluate the generated summaries. The experimental results demonstrate that the AAF method proposed in this article achieved significant improvements in the BERTScore metric, with precision, recall, and F1 score increasing by 7%, 12%, and 9%, respectively, compared to the baseline model. This result further validates the effectiveness of the AAF method in semantic preservation, with detailed data shown in Table 5.

The study will also conduct a qualitative analysis from three dimensions: coherence, informativeness, and fluency. The five-point Likert scale is adopted as the evaluation tool to quantify the subjective judgments of evaluators on the generated summaries. The specific definitions of the three evaluation dimensions are as follows:

- **Coherence:** Refers to the logical consistency within the summary, including whether the sentences are naturally connected and whether the theme is clear and free of contradictions.

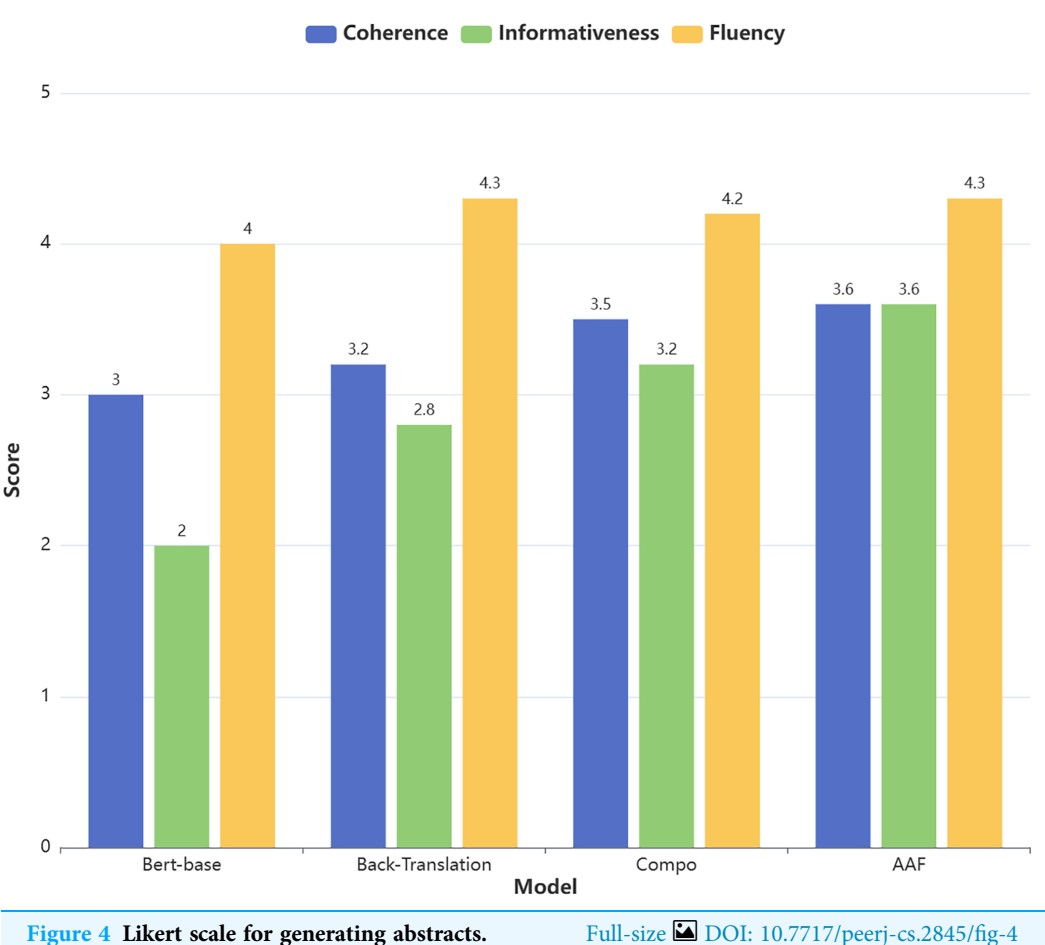

**Figure 4  Likert scale for generating abstracts.**     

- **Informativeness:** Refers to whether the summary adequately captures and conveys the core information of the original text, including key facts or viewpoints.
- **Fluency:** Refers to whether the language expression of the summary is natural, grammatically correct, and easy to read and understand.

   This study invited three evaluators with relevant domain expertise to participate in the assessment. Each evaluator independently scored 30 generated summaries in a distraction-free environment. The result is shown in Fig. 4 below.

   The evaluation results show that the summaries generated using AAF data augmentation performed best in the fluency dimension (average score of 4.3), while their performance in the informativeness dimension was relatively weaker (average score of 3.6). Specifically, the summaries generated with AAF data augmentation were able to adequately capture the core information of the original text. However, since the evaluation process relies on subjective judgments, future research could incorporate automated evaluation methods to further validate the results.

# PARAMETER ANALYSIS

## Analysis of augmented data proportions

During each model training process, even though we only doubled the amount of training data from the original 1%, we still observed fluctuations in the data during the later stages of training. This indicates that while increasing the amount of augmented data may improve the overall data volume to some extent, the model may still overly rely on the augmented data, leading to overfitting. This prompts a further exploration of the impact of augmented data quantity on model performance.

In the validation experiments, we gradually reduced the proportion of augmented data used to observe its effect on model performance. Specifically, the usage ratio of augmented data was systematically decreased from an initial 100%, with the usage ratio $\alpha$ set between 0% and 100% to control the amount of augmented data utilized. In this experiment, we did not randomly select augmented data; instead, we respectively filtered out data from the datasets generated by the two methods. The amount of augmented data produced by the MPA method is denoted as $X_P$, while the data generated by the SRA is denoted as $X_R$. The initial total amount of training data is denoted as $x$, and the calculations are as shown in the following formula:

$$X_P = \alpha(0.5 - \lambda)x \tag{8}$$
$$X_R = \alpha(0.5 + \lambda)x \tag{9}$$
$$X_{\text{total}} = X_P + X_R. \tag{10}$$

To simplify the experimental operations, we take the values of $\alpha$ as 10%, 30%, 50%, 70%, 90%, and 100%. The AAF method was applied to the DialogSum dataset, and two validation experiments were conducted with augmentation fusion coefficient adjustments $\Delta\lambda$ set at 0.05 and 0.1. The results are presented in Table 6.

The experimental results indicate that the AAF has a significant impact on model training performance when using different proportions of augmented data. In both sets of experimental parameters, the optimal results were achieved when the extraction ratio was set at 50%. These further underscores the importance of rationally configuring the quantity of augmented data for model performance. It serves as a reminder that incorporating excessive augmented data in future experiments may not only fail to improve model performance but could also lead to a decline in performance due to the introduction of excessive noise. This situation is particularly pronounced when data resources are limited or when dealing with noisy datasets.

## Analysis of augmentation fusion coefficient adjustment $\Delta\lambda$

In the AAF, the augmentation fusion coefficient is used to adjust the ratio between the two augmentation methods, directly affecting the model's training process and final performance. As a result, the value of its adjustment $\Delta\lambda$ is an important parameter. If a

**Table 6 Summary performance comparison using different proportions of augmented data.**

| $\Delta\lambda$ | $\alpha$ | ROUGE-1 | ROUGE-2 | ROUGE-L |
|---|---|---|---|---|
| 0.05 | 10% | 40.27 | 13.77 | 31.53 |
| | 30% | 40.62 | 14.69 | 32.67 |
| | 50% | 40.95 | 14.69 | 32.83 |
| | 70% | 40.70 | 14.41 | 32.41 |
| | 90% | 40.68 | 14.84 | 32.70 |
| | 100% | 40.81 | 15.16 | 32.67 |
| 0.1 | 10% | 41.02 | 14.64 | 33.04 |
| | 30% | 41.08 | 15.30 | 32.94 |
| | 50% | 41.38 | 15.09 | 33.52 |
| | 70% | 41.16 | 14.86 | 32.84 |
| | 90% | 40.87 | 15.19 | 32.79 |
| | 100% | 41.02 | 14.61 | 32.58 |

**Table 7 Relationship between the values of $\Delta\lambda$ and model performance.**

| $\Delta\lambda$ | ROUGE-1 | ROUGE-2 | ROUGE-L |
|---|---|---|---|
| 0.05 | 40.95 | 14.69 | 32.83 |
| 0.1 | 41.38 | 15.09 | 33.52 |
| 0.15 | 40.73 | 14.86 | 32.57 |
| 0.2 | 40.57 | 14.31 | 32.26 |

large adjustment is made at once, such as setting a substantial change in the fusion coefficient, it may lead to difficulties for the model in quickly adapting to the new data distribution and failing to fully learn the characteristics of the new data, resulting in a short-term decline in performance. This is especially true when training on small datasets, where it may even cause the training process to diverge. Based on our observations in the experiments, gradually and slowly adjusting the distribution and methods of augmented data is more beneficial for the model to adapt smoothly to new data. This refined adjustment strategy helps reduce the risk of overfitting, as the model gradually accepts new data, allowing for more effective weight updates. Compared to abrupt adjustments, a gradual adjustment process enables the model to more naturally engage with different augmentation methods, thereby enhancing its generalization capability and ultimately leading to better performance on real data distributions.

To further validate this observation, we performed comparative experiments using four distinct values for the augmentation fusion coefficient: 0.05, 0.1, 0.15, and 0.2. The experimental results are shown in Table 7. By varying the magnitude of the adjustment to the fusion coefficient, we observed the effects of different variable sizes during the model training process. Notably, when the fusion coefficient was set at 0.1, the model achieved the

best performance. This experimental data provides a more precise basis for optimizing the augmentation strategy.

## CONCLUSION

In this study, we propose a data augmentation method based on AAF for abstract dialogue summary. This method aims to address the problem of insufficient training data. This method analyzes the advantages and disadvantages of simple character-level MPA and generation-based SRA. Through the adaptive augmentation fusion strategy, it effectively balances the influence of both augmentation methods on the model's learning performance and generalization capability. Experimental results demonstrate that, under resource constraints, the AAF significantly outperforms individual augmentation methods in dialogue summary tasks. Future research will further optimize and expand the potential of the AAF in other domains, such as incorporating data augmentation techniques that have shown exceptional performance in areas like meeting summaries (*Feng et al., 2020*), and integrating them with the adaptive augmentation fusion strategy, with the aim of broadening the application of this augmentation fusion method to more extensive practical fields.

### Funding

This work was supported by the Beijing Natural Science Foundation-Beijing Municipal Education Commission Joint Project (KZ202010015021), the National Natural Science Foundation of China project (10000200223), the Research Program of Beijing Municipal Education Commission (KM202110015003, KM202410015002), the Youth Support Program of Beijing Institute of Graphic Communication (20190125031) and the Doctoral Research Startup Fund of Beijing Institute of Graphic Communication (27170124010). The funders had no role in study design, data collection and analysis, decision to publish, or preparation of the manuscript.

### Grant Disclosures

The following grant information was disclosed by the authors:
Beijing Natural Science Foundation-Beijing Municipal Education Commission Joint Project: KZ202010015021.
National Natural Science Foundation of China Project: 10000200223.
Research Program of Beijing Municipal Education Commission: KM202110015003, KM202410015002.
Youth Support Program of Beijing Institute of Graphic Communication: 20190125031.
Doctoral Research Startup Fund of Beijing Institute of Graphic Communication: 27170124010.

### Competing Interests

The authors declare that they have no competing interests.

## Author Contributions

- Weihao Li conceived and designed the experiments, performed the experiments, analyzed the data, performed the computation work, prepared figures and/or tables, authored or reviewed drafts of the article, and approved the final draft.
- Dan Jiang conceived and designed the experiments, authored or reviewed drafts of the article, funding, and approved the final draft.
- Han Zhang conceived and designed the experiments, authored or reviewed drafts of the article, funding, and approved the final draft.
- Kejing Xiao conceived and designed the experiments, authored or reviewed drafts of the article, funding, and approved the final draft.
- Shaozhong Cao conceived and designed the experiments, authored or reviewed drafts of the article, funding, and approved the final draft.

## Data Availability

The code is available at GitHub and Zenodo:
- https://github.com/alolke/AAF.
- alolke. (2025). alolke/AAF: Source code and dataset for AAF (alolke/AAF). Zenodo. https://doi.org/10.5281/zenodo.14975390.

The third-party datasets at available at Hugging Face:
- DialogSum: https://huggingface.co/datasets/knkarthick/dialogsum.
- SAMSum: https://huggingface.co/datasets/Samsung/samsum.

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
