# Peer review of "An adaptive fusion-based data augmentation method for abstract dialogue summarization"

_PeerJ Computer Science, doi:10.7717/peerj-cs.2845_

## Round 0.1 · original submission · Major Revisions

You should respond in detail to the comments of both expert reviewers

Reviewer 1 ·

Basic reporting

All comments have been added in detail to the last section.

Experimental design

All comments have been added in detail to the last section.

Validity of the findings

All comments have been added in detail to the last section.

Additional comments

Review Report for PeerJ Computer Science
(An adaptive fusion-based data augmentation method for abstract dialogue summarization)

1. Within the scope of the study, an adaptive augmentation fusion method called AAF was performed using different datasets related to abstract dialogue summarization.

2. In the introduction section, the importance of the subject, the data scarcity problem and the challenges related to the subject were mentioned. Accordingly, the main contributions of the study were mentioned. However, it would be useful to detail the contributions in this section.

3. In the literature review section, data augmentation methods such as adversarial data and token-level in natural language processing were mentioned. In addition, the literature on abstract dialogue summarization was also mentioned. The literature review in this section definitely needs to be detailed. A detailed literature table summarizing important parts such as pros and cons and originality points can be added.

4. When the adaptive augmentation fusion method, diagram and parameter settings proposed in the study are examined in detail and compared with the literature, it is observed that it contains a certain level of originality and has the potential to contribute to the literature.

5. For the dataset required for using and testing the proposed method, two different datasets, DialogSum and SAMSum, were preferred in the study. Using more than one dataset in the study, instead of sticking to a single dataset, increased the quality of the study. However, in this section, it should be stated more clearly why these two datasets were preferred compared to the dataset in the literature and/or whether experiments were conducted with different datasets.

6. When the evaluation metric types and results are examined, it is understood that they are at a sufficient and appropriate level in terms of literature.

7. Sharing the codes related to the study as open source on the github platform will increase the trust in the study and make the study more usable after publication, thus increasing its contribution to the literature.

As a result, the study proposes an important data augmentation method for abstract dialogue summarization. However, paying attention to the above sections will take the quality of the study to the next level.

Reviewer 2 ·

Basic reporting

The manuscript is generally well-written and presents an innovative method, Adaptive Augmentation Fusion (AAF), for data augmentation in dialogue summarization. However, several areas require improvement to enhance clarity, rigor, and completeness. Below is a detailed review:

Language and Grammar
While the language is professional, there are instances of grammatical errors and formatting issues that should be addressed:
- Correct the word "Reduceing" in L146 to "reducing."
- In Figure 1, correct the word "Replaace" with "Replace", and add a space before (AAF) in the caption.
- Correct the number of training examples of SAMSum in Table 1 from "147327" to "14,732".
- The first mention of AAF (Adaptive Augmentation Fusion) in L52 should include an explicit description of the acronym.
- Ensure that appropriate spaces are added before introducing the acronyms MPA and SRA in L52–53 for better readability.
- Add a space after the citation in L72 to maintain formatting consistency.
Recommendation: Conduct a thorough proofreading of the manuscript to eliminate all minor grammatical errors and ensure a smooth reading experience.

Motivation for Data Augmentation:
The manuscript highlights the importance of data augmentation, but the motivation is not sufficiently justified. In L37, the claim that 16,369 labeled summaries of SAMSum are few is unconvincing. In practice, real-world low-resource scenarios often involve datasets with as few as 100 labeled examples, which is an order of magnitude smaller than the datasets referenced in this study.
Recommendation: Revise the motivation to focus on the challenges of true low-resource regimes with minimal labeled data. Highlight how AAF can be effective in such scenarios, emphasizing its potential application in domains with severely limited annotations.

Related Work:
The Related Work section provides a comprehensive overview of various data augmentation methods in NLP, including token-level, sentence-level, adversarial, and hidden-space augmentation techniques. However, it omits a discussion of segmentation-based augmentation, which is particularly effective for summarization tasks involving long input documents. Segmentation-based augmentation is a critical method that divides long inputs into smaller chunks and assigns relevant portions of the gold summary to each chunk. This implicitly augments the training data, reduces input complexity, and enhances coherence without relying on external augmentation techniques.
Recommendation: To strengthen the manuscript, the authors should add this category of data augmentation to the Related Work section.

Experimental design

The manuscript addresses the important issue of data scarcity in abstractive dialogue summarization. It clearly identifies the gap and proposes a novel adaptive augmentation strategy. The approach is original.

Below are specific suggestions for improvement:
- To improve the presentation of results in Table 3, make the highest results bold and the second-best underlined.
- The manuscript does not explicitly state how many training examples are generated after data augmentation. This could be a critical omission, as it directly impacts the interpretation of results. After applying AAF, what is the final size of the training dataset for both DialogSum and SAMSum?

Validity of the findings

The manuscript effectively uses ROUGE scores to evaluate the performance of the proposed method, which is appropriate for assessing lexical overlap between the generated and reference summaries. However, additional metrics could provide a more comprehensive evaluation of the model's performance:

Quantitative Metrics:
Incorporating BERTScore or BARTScore would allow for a deeper semantic assessment of the generated summaries. Unlike ROUGE, these metrics evaluate the contextual and semantic alignment of the generated text with the reference text, which is crucial for abstractive summarization tasks like dialogue summarization.

Qualitative Metrics:
A human evaluation of the generated summaries would significantly enhance the evaluation by addressing aspects like coherence, informativeness, and fluency that automated metrics cannot fully capture.
I recommend performing the evaluation with a small sample size (e.g., 50 summaries) rated on a Likert scale for the following key dimensions:
Coherence: Logical flow of ideas.
Fluency: Grammatical and syntactic correctness.
Informativeness: Degree to which the generated summary captures the key points of the input.

---

## Round 0.2 · accepted · Accept

Dear Authors,

Thank you for addressing the reviewers' comments and concerns. Your manuscript now seems sufficiently improved and ready for publication.

Best wishes,

Reviewer 1 ·

Basic reporting

All comments have been thoroughly added in the final section.

Experimental design

All comments have been thoroughly added in the final section.

Validity of the findings

All comments have been thoroughly added in the final section.

Additional comments

Review Report for PeerJ Computer Science
(An adaptive fusion-based data augmentation method for abstract dialogue summarization)

Thank you for the revision. The changes in the paper are sufficient, and the responses to the reviewers' comments are appropriate. Therefore, I recommend that the paper be accepted. I wish the authors success in their future research. Best regards.

Reviewer 2 ·

Basic reporting

N/A. See Additional comments.

Experimental design

N/A. See Additional comments.

Validity of the findings

N/A. See Additional comments.

Additional comments

I appreciate the authors' revisions and improvements to the manuscript. The writing is now clearer and more refined. The authors have successfully incorporated many of my suggestions, enhancing the overall quality of the paper.

However, the newly added related work section on data augmentation techniques remains somewhat limited, with relatively few new references to previous studies. For instance, methods such as Se3 (introduced in "Semantic Self-Segmentation for Abstractive Summarization of Long Documents in Low-Resource Regimes") and Athena (proposed in "Align-Then-Abstract Representation Learning for Low-Resource Summarization") were explicitly designed to address data augmentation for text summarization using a segmentation-based approach. Since the paper focuses on data augmentation for a summarization task, the authors should incorporate these and other relevant works to better position their study and further strengthen the discussion.